# Different Oviposition Strategies of Closely Related Damselfly Species as an Effective Defense against Parasitoids

**DOI:** 10.3390/insects10010026

**Published:** 2019-01-10

**Authors:** Filip Harabiš, Tereza Rusková, Aleš Dolný

**Affiliations:** 1Department of Ecology, Faculty of Environmental Sciences, Czech University of Life Sciences Prague, Kamýcká 129, CZ-165 00 Praha–Suchdol, Czech Republic; harabis@fzp.czu.cz; 2Department of Biology and Ecology, Faculty of Science, University of Ostrava, Chittussiho 10, CZ-710 00 Slezská Ostrava, Czech Republic; Snapeterka@seznam.cz

**Keywords:** oviposition strategy, cost of reproduction, egg mortality, trade-off, Odonata, damselfly life history, egg parasitoid

## Abstract

Parasitoidism is one of the main causes of insect egg mortality. Parasitoids are often able to detect eggs using semiochemicals released from eggs and disturbed plants. In response, female insects adopt a wide variety of oviposition strategies to reduce the detectability of eggs and subsequent mortality. We evaluated the proportion of parasitized and undeveloped eggs of three common damselfly species from the family Lestidae, the most diverse group of European damselflies, in terms of oviposition strategies, notably clutch patterning and the ability to utilize oviposition substrates with different mechanical properties. We assumed that higher costs associated with some oviposition strategies will be balanced by lower egg mortality. We found that the ability of *Chalcolestes viridis* to oviposit into very stiff substrates brings benefit in the form of a significantly lower rate of parasitoidism and lower proportion of undeveloped eggs. The fundamentally different phenology of *Sympecma fusca* and/or their ability to utilize dead plants as oviposition substrate resulted in eggs that were completely free of parasitoids. Our results indicated that ovipositing into substrates that are unsuitable for most damselfly species significantly reduces egg mortality. Notably, none of these oviposition strategies would work unless combined with other adaptations, such as prolonging the duration of the prolarval life stage or the ability to oviposit into stiff tissue.

## 1. Introduction

The selection of optimal oviposition sites is a key option to increase fitness, especially in animals who do not provide parental care to their offspring. Thus, this applies to odonates, in which oviposition site selection can significantly affect egg mortality [1,2]. The insertion of eggs into living or decaying plant tissues, referred to as endophytic oviposition, reduces water loss, minimizes winter mortality, and provides some protection from natural enemies [3]. Most damselflies and some dragonflies (family Aeshnidae) use this endophytic oviposition strategy, whereas most other dragonfly species employ exophytic oviposition [4,5]. 

Endophytic oviposition is an ancient strategy for egg laying in Odonatoptera; it is documented from the Triassic, but its origin may be even older [6,7]. The endophytic ovipositor of damselflies consists of three pairs of valvulae to facilitate oviposition into substrates, such as plant tissues [8,9]. Most damselfly species use a wide range of host plants, but some species prefer particular plant species, or particular parts of plants, to lay their eggs in. Sensilla on the valves and styli may be involved in oviposition plant recognition [10,11,12]. Female damselflies in the family Lestidae (e.g., *Lestes* spp.), in most cases, utilize plants that are submerged and often have relatively soft tissues; however, some species (e.g., *Chalcolestes* spp.) prefer to oviposit into stiff tissues, such as reeds or even tree bark [9,13]. Additionally, some other lestids (e.g., *Sympecma* spp.) oviposit into decaying plant material [12,14].

Based on current knowledge, Lestidae is the most diverse group of European damselflies that exhibit clutch patterning [13]. Matushkina et al. [9] showed that a high diversity in clutch patterning was related to the mechanical properties of the host plant and their availability. However, several species, such as *Chalcolestes viridis*, prefer stiff substrates, despite the availability of softer plants (personal observation). In general, lestid females need more time to deposit an egg compared to other damselflies [8,13], which may result in higher mortality risk during oviposition [15,16]. However, they have a solid ovipositor and are able to exert the necessary force for oviposition into hard plant tissues [8]. Thus, there is a trade-off between the cost of searching for oviposition sites, including mortality risk during oviposition, and the ability to hide/protect eggs. This is also an example of the traditional trade-off between quantity and quality (i.e., between offspring number and parental investment in their protection). In general, odonates with endophytic oviposition deposit less than ∼300 eggs at one oviposition event, while exophytic odonates often lay 1500 or more eggs in each clutch [4,5]. Egg mortality is widespread in aquatic insects; therefore, females developed various strategies to defend their eggs against predators. The efficiency of such defense is related to the search capabilities of the predator; therefore, oviposition in hard-to-access or cryptic locations may be an effective strategy. Eggs may be defended physically and chemically [17]. 

It is known that egg parasitoids are able to use the mixtures of plant volatiles (semiochemicals induced during host oviposition) and host odour for host localization [18,19]. Significantly more eggs are parasitized when they are clumped [20,21]. Additionally, they are significantly less parasitized when they are oviposited on habitat edges than the eggs oviposited in the center of the habitat [21]. This indicates that investment in oviposition site selection may greatly affect offspring fitness. Freshwater insects have a wide range of oviposition strategies to reduce egg detection by egg parasitoids [22,23,24]. As already mentioned, the distribution of lestid eggs is not random: females show a clear preference for certain host plants and even for specific microhabitats, and eggs are laid in multi-egg clumps [12,25]. However, very little is known about how females’ preference for specific oviposition substrates influences the likelihood of egg mortality. In this study, we compared the proportion of parasitized eggs of different lestid species with different oviposition strategies, focusing on their preference for specific oviposition substrate (i.e., plant species). 

We evaluated two main hypotheses:There are differences in protection against egg parasitoids between different study species. Preference for stiff substrate in *C. viridis* can provide significant protection from egg parasitoids.There are differences in overall egg mortality rates between individual damselfly species.

## 2. Methods

### 2.1. Biology of Study Species and Their Oviposition Strategies

All studied species are common in Central Europe and occur in a wide range of lentic habitats. Their occurrence is mainly limited by the availability of suitable oviposition sites. The preferred oviposition substrate of individual species has many different mechanical properties, which could have a significant impact on egg clutch patterning [15].

*Lestes sponsa* (Hansemann, 1823) is widely distributed in the Palearctic, apart from the southern and northern extremes. The emergence of *L. sponsa* starts in Central Europe at the end of May, the main activity period occurs in July and August, and the last individual is usually seen at the end of September [26]. Oviposition proceeds immediately after copulation and usually occurs while males remain attached to the females [14]. *L. sponsa* typically places its eggs into the stalks of emergent plants (e.g., *Juncus*, *Equisetum*, *Schoenoplectus*, or *Eleocharis* spp.), but also into halophyte leaves (e.g., *Iris* or *Stratiotes* spp.) [27]. The female first penetrates the plant tissue with her ovipositor and then usually lays four eggs in the groove [13]. Females of *L. sponsa* form complex linear clutches, with perforations arranged in single linear rows oriented along the fibers of the plant tissue (Table 1).

*Chalcolestes viridis* (Vander Linden, 1825) is a common species in the Western Palaearctic, except for the eastern and northern parts of Europe and the whole of Asia [28]. The emergence of *C. viridis* starts in Central Europe in mid-June, the main activity period occurs in August, and the last individual is usually seen at the end of October [26]. Oviposition proceeds immediately after copulation, in tandem [14]. *C. viridis* usually places its eggs into stiff substrates, primarily the bark of softwood trees (e.g., *Salix*, *Populus*, or *Alnus* spp.). Female *C. viridis* have to expend great energy to penetrate the stiff substrate; therefore, their oviposition rate is very slow compared to other damselfly species [13]. Egg clutch patterning is probably related to the oviposition substrate (Table 1). Eggs are oviposited in complex linear clutches, with eight eggs in one perforation. Perforations are arranged in two linear rows [13]. 

*Sympecma fusca* (Vander Linden, 1820) is a common species with a similar distribution to *C. viridis*. However, the origin of the species extends to semi-desert areas of Central Asia [28]. The area of origin is related to the very different phenology of this species (genus) compared with those of the other European lestids. The emergence of *S. fusca* starts in Central Europe in the end of July. After a few days, individuals leave the aquatic habitat and return in early April or before [26]. Meanwhile, the pre-adults survive by overwintering in terrestrial habitats [29]. At the beginning of spring, mature individuals return to aquatic habitats to oviposit in tandem into the floating vegetation. The eggs are oviposited in a *Coenagrion*-type patterning with simple zigzag-like clutches, with one egg per perforation [15]. In comparison with other lestids, *S. fusca* oviposition is very fast, since eggs are usually oviposited into soft decaying plant material [12] (Table 1).

### 2.2. Study Sites 

Eight locations with the occurrence of all three species were situated in the foothills of the Beskid Mountains in the northeastern Czech Republic (Table 2). Four of the study sites were extensively managed ponds and four were artificial pools with rich littoral and floating vegetation. The climate of the study area is characterized by long, warm, and moderately dry summers, warm springs, and slightly warm autumns. Winter is short, warm, and very dry, with a short duration of snow cover. The average annual temperature is 6–7 °C and average annual rainfall is 650–750 mm.

### 2.3. Data Sampling

Sampling was carried out during species-specific periods of high oviposition activity. The sampling period for overwintering *S. fusca* was between 23 April and 14 May 2016, and for the late summer species, *L. sponsa* and *C. viridis*, was between 4 and 25 July 2016. Twenty plants with numerous oviposition scars characteristic of each species were collected at each site. Individual plants were at least two meters from each other. Immediately after sampling, plants were placed in the cold to delay parasitoid and egg development. Subsequently, all eggs were excised from the plant tissues and the overall numbers of parasitized and undeveloped eggs were evaluated using dissecting microscope at magnitude 40×. The development of damselfly eggs and parasitoids before cooling was fast and differences in morphological changes of eggs containing parasitoid were clearly visible. Unfertilized or damaged eggs were categorized as undeveloped eggs. For the details and photo-documentation see [24].

We analyzed the seasonal change in the proportion of parasitized and undeveloped eggs. We found that this proportion did not change significantly in any species during the sampling period (three weeks); therefore, seasonal effect was not included in subsequent models. Confusion of *L. sponsa* eggs with the eggs of other species of the genus was possible. However, regarding the dominant species and the characteristic patterning of eggs in individual species, it was very unlikely. Parasitoids and oviposition (plant) substrate was determined to genus level. Damselfly nomenclature followed [30].

### 2.4. Statistical Analysis

We analyzed differences in the proportion of parasitized eggs and egg development success among damselfly species (hosts). Additionally, we tested the influence of the oviposition plant on both the proportion of parasitized eggs and development success. Differences between localities were considered by employing location as a random effect. Therefore, we used a generalized mixed models (GLMM) with a binomial distribution of errors in the R package lme4 [31]. Damselfly species and oviposition plants were variables with fixed effect. When we analyzed the abundance of different parasitoid species extirpated from the damselfly eggs, we used a GLMM with negative binomial distribution of errors, where parasitoid abundance was the response variable, and parasitoid species and damselfly species (host) were fixed effects. Location was also used as a random effect. Because we had different numbers of eggs of individual damselfly species, we first converted the number of parasitoids per 1000 host eggs. Multiple comparisons of means (using Tukey contrasts) in the R package multcomp [32] were used to test differences between damselfly species. Statistical significance was established using α = 0.05. All analyses were performed in R 3.2.2 [33].

## 3. Results

In total, 28,098 eggs of all lestid species were extracted from plant tissues (Table 3). There was a significantly higher proportion of parasitized eggs for *L. sponsa* (4.7%) than for *C. viridis* (1.0%), whereas we did not find any parasitoids in the eggs of *S. fusca* (Table 3 and Table 4, Figure 1) while the proportion of parasitized eggs was not affected by oviposition plant (Table 4). We also looked at the proportion and abundance of individual species of parasitoids. Interestingly, parasitoid species differed significantly between eggs of damselfly species (Table 5, Figure 2). According to our results, the prevalence of parasitoids from the genus *Aprostocetus* was significantly higher in *L. sponsa* than in *C. viridis*, whereas no such trend was found for *Prestwichia aquatica* (Figure 2). The proportion of undeveloped eggs significantly varied across species and was not affected by oviposition plant (Figure 3, Table 6). The capability of *C. viridis* to oviposit into stiff tissues, outside the aquatic habitat, provided other additional benefits, such as a lower proportion of undeveloped eggs compared to the other species. A significantly higher proportion of undeveloped eggs was found in *L. sponsa* (Figure 3). 

## 4. Discussion

Species from the family Lestidae represent damselflies with very diverse egg-laying strategies, differing in clutch size, host plant, and microclimate [13]. Based on our findings, it is evident that eggs of species with different oviposition preferences have a significantly different survival outcome. It seems that the higher costs associated with some oviposition strategies could be considered as a trade-off between investment into costly oviposition strategy and parasitoid avoidance.

The level of parasitoid protection in eggs laid into soft plant tissues in *L. sponsa* was lower than that in *C. viridis* eggs oviposited into stiff plants tissues. This indicates that the ability of *C. viridis* to oviposit into substrates such as tree bark provides a defense for the eggs and increases the chances of offspring survival, whereas the *L. sponsa* eggs deposited into the soft-stemmed herbs are more exposed to enemies. Species that invest heavily in egg defense are unable to lay as many eggs as species that do not protect their eggs [34] like our study species. 

A significantly lower proportion of undeveloped eggs in *C. viridis* may be due to the fact that their overall oviposition rate is very low compared to other damselflies [13]. Females produce larger clusters of eggs, which is a “logical choice” due to the stiffness of their preferred oviposition substrates. However, prolonged oviposition means that *C. viridis* females are more susceptible to predation [5]. In this respect, oviposition of *S. fusca* into soft plant tissues may be the most advantageous strategy. Oviposition rates of *S. fusca* are significantly higher than in other lestids (except *L. barbarus*) [13] and have a relatively low proportion of undeveloped eggs. Another potential advantage may be the fact that eggs placed in floating dead plant tissues are very difficult for non-aquatic predators to reach. Furthermore, dead plants do not produce semiochemicals, which can be used for egg detection. This may be an explanation why *S. fusca* eggs were completely free of parasitoids. However, dead plant tissues have only a short-term durability. Therefore, eggs placed in this substrate must hatch promptly after oviposition and may not survive a long overwintering period. Thus, dead plant material, as an oviposition substrate, would be suitable for a limited number of species, notably those who will not overwinter in plant tissues. Specific life history of damselflies in the genus *Sympecma* includes several unique adaptations, including adult overwintering and rapid larval development [14]. The whole complex of adaptations yields benefits by reducing competition for both larvae and adults [35]. According to our results, phenological shift, obtained through adult overwintering, may be another reason why parasitoids were absent from *S. fusca* eggs. During coevolution, parasitoids with short-lived adult life stages tend to focus their development to a period when sufficient hosts (eggs) are available [36]. Accordingly, there could be an enemy-free space for species with fundamentally different phenology, such as *S. fusca*. 

Unique adaptations of the genus *Sympecma*, including phenological shift, evolved in extreme arid conditions [14,37]. Other lestid species must protect their eggs against parasitoids. Based on our findings, the most common parasitoid species *Aprostocetus* (*Ootetrastichus*) *pseudopodiellus* was significantly less abundant in *C. viridis* eggs than in those of *L. sponsa*. This indicates that at least several parasitoid species are somehow limited in their ability to detect and/or attack *C. viridis* eggs. We assume this is because *C. viridis* eggs, which are generally oviposited into stiff tissues, are better protected against predation. Another possible explanation is that *C. viridis* tends to oviposit into the bark of trees, which are usually several meters away from the aquatic habitat, and can thus be at the edge of interest for parasitoids looking for aquatic hosts. This concurs with Cronin et al. [21], who have found significantly fewer parasitoids at the habitat edges.

Conversely, while oviposition away from the aquatic habitat may be beneficial for eggs, it could pose an insurmountable problem for larvae. Immediately after hatching, larvae must find water. Therefore, in only several odonate species, such as *C. viridis*, the first instar is a “true prolarva” that can reach the water by jumping or springing [5]. In most odonate species, the prolarval stadium is extensively reduced and lasts less than one minute [4,38]. The mortality of prolarvae that must cross land to reach water is probably high, and therefore should not be disregarded in estimates of the overall survival of individual species [5].

## 5. Conclusions

In conclusion, our findings show that egg mortality varies significantly between species. According to our results, it is obvious that the ability to lay eggs into plant substrates that are somewhat unsuitable for oviposition (e.g., stiff or decaying plant tissues), and thus are not used as often, may serve as an efficient strategy against some parasitoid groups. Although the mechanisms of the egg search in odonate parasitoids are poorly known, our results suggest that oviposition into different substrates may elicit different chemical reaction, which may consequently affect the ability to detect host eggs. Moreover, a suitable oviposition substrate can significantly enhance survival during overwintering, by mitigating the effect of some lethal environmental processes (e.g., the formation of ice crystals or desiccation). Broadly, however, the utilization of a particular substrate is not only related to the mechanical properties of the plant tissues and female oviposition capabilities, but also to other adaptations, such as prolonged prolarva in *C. viridis* or adult overwintering in *S. fusca*. 

## Figures and Tables

**Figure 1 insects-10-00026-f001:**
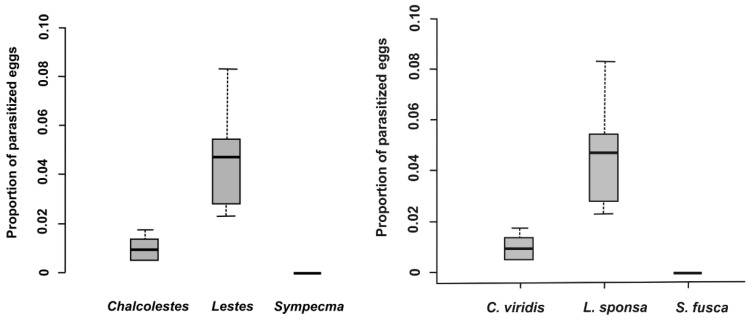
Proportion of parasitized eggs per damselfly species.

**Figure 2 insects-10-00026-f002:**
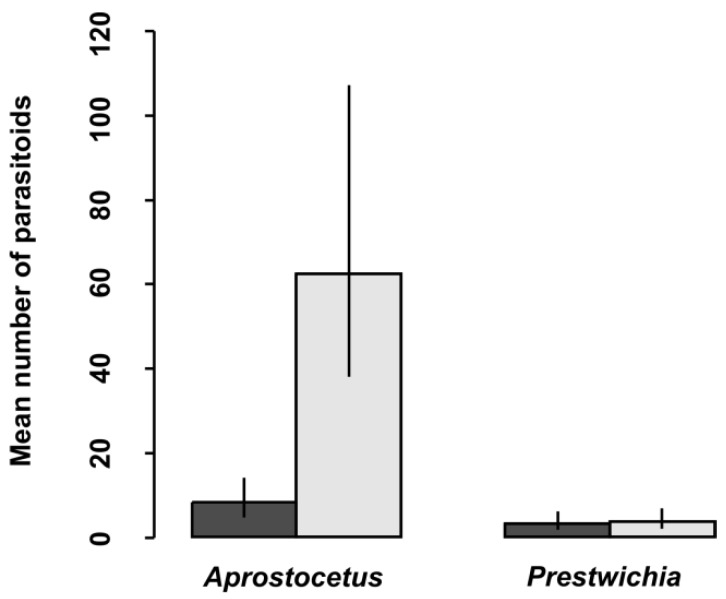
Mean number of parasitoids in *Lestes sponsa* (light) and *Chalcolestes viridis* (dark) eggs, with 95% confidence intervals.

**Figure 3 insects-10-00026-f003:**
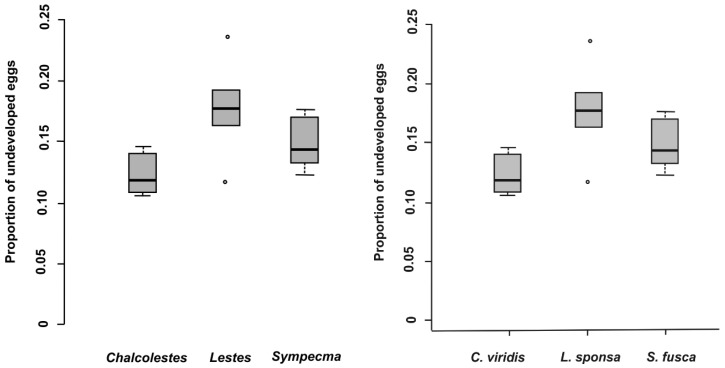
Proportion of undeveloped eggs per damselfly species.

**Table 1 insects-10-00026-t001:** Egg clutch patterning and mechanical properties of the endophytic ovipositor in the studied species (Matushkina and Gorb, 2002, 2007).

Parameter	Units	*Lestes sponsa*	*Chalcolestes viridis*	*Sympecma fusca*
perforation arrangement		one linear row	two linear rows	zigzag
eggs per perforation		1–4	1–8	1
oviposition rate *	eggs/min	2.5	1.0	3.3–5.0
ovipositor stiffness *	N/m	328 ± 26	409 ± 11	192 ± 11
oviposition plant in our study		*Juncus* sp., *Equisetum* sp.	*Salix* sp., *Alnus* sp.	*Typha* sp., *Eleocharis* sp.

* Several parameters were not available directly for our study species; therefore, we stated values for the closely related species *Sympecma paedisca* and *Chalcolestes parvidens*.

**Table 2 insects-10-00026-t002:** Location of the study sites.

Locality	Lat.	Long.	Habitat Type
Loc_1	49.81389	18.4456722	artificial habitat
Loc_2	49.78697	18.4066994	extensive pond
Loc_3	49.87960	18.1750889	extensive pond
Loc_4	49.82811	18.5013144	artificial habitat
Loc_5	49.83172	18.5042417	artificial habitat
Loc_6	49.58854	18.1264561	artificial habitat
Loc_7	49.63516	18.1014001	extensive pond
Loc_8	49.63073	18.1032972	extensive pond

**Table 3 insects-10-00026-t003:** The total number of lestid eggs excised from plant tissues and the proportion of undeveloped/parasitized eggs.

Host	Undeveloped	% Mean	Parasitized	% Mean	Total Eggs
*Lestes sponsa*	1258	17.6%	417	4.7%	8040
*Chalcolestes viridis*	1833	12.2%	132	1.0%	14,778
*Sympecma fusca*	784	14.8%	0	0.0%	5280
∑	3875		549		28,098

**Table 4 insects-10-00026-t004:** Generalized mixed model indicating differences in parasitoidism of lestid eggs in relation to host species and oviposition plant.

Model	df	AIC	Chisq	*P*
~ (1 | Locality)	2	721.5		
~ Host + (1 | Locality)	2	514.9	210.57	<0.001
~ Host + Plant + (1 | Locality)	6	523.3	3.66	0.723

**Table 5 insects-10-00026-t005:** Effect of parasitoid species and damselfly species (host) on parasitoid abundance.

Model	df	AIC	Chisq	*P*
~ (1 | Locality)	3	190.0		
~ Host + (1 | Locality)	1	182.8	9.3	0.002
~ Host + Parasitoid + (1 | Locality)	1	169.6	15.2	<0.001
~ Host + Parasitoid + Parasitoid:Host + (1 | Locality)	1	163.0	8.5	0.003

**Table 6 insects-10-00026-t006:** Generalized mixed model indicating differences in relative proportion of undeveloped lestid eggs in relation to host species and oviposition plant.

Model	df	AIC	Chisq	*P*
~ (1 | Locality)	2	1459.9		
~ Host + (1 | Locality)	2	1423.4	40.53	<0.001
~ Host + Plant + (1 | Locality)	6	1424.7	10.74	0.097

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
