# Peer review of "Different Oviposition Strategies of Closely Related Damselfly Species as an Effective Defense against Parasitoids"

_insects, 2019, doi:10.3390/insects10010026_

Reviewer 1 Report

In their manuscript Different oviposition strategies of closely related damselfly species as an effective defense against parasitoids, the  authors compared the proportion of parasitized eggs, and egg-development, among three species of damselfy host species. This is a nice work that with some minor changes will produce a very solid piece of work.

General comment:

authors should keep an uniform nomenclature for the species throughout the text and the figure legends (e.g. L. sponsa, C. viridis and S. fusca in the text vs Lestes, Chalcolestes and Sympecma in Figure legends)

Specific comments:

lines 73 to 74: the sentence The proposed research would contribute to our knowledge of the species-specific life history traits and their influence on defense against parasitoids reads as copied from a grant application. Authors should shift the sentence to the Discussion section (e.g. these results/this study contributes to our knowledge...')

lines 77 to 84: idem comment above. If hypotheses are to be mentioned in the introduction, authors must include a reference to the validation/refusal of such hypotheses in the discussion.

lines 86 to 123: the information contained in the section Biology of studies species and their oviposition strategies is not mere methodological, thus authors should consider including this complete section into results.

line 124: authors should move Table 2 to the end of Study sites section (below line 132)

lines 133 to 146 - Data sampling section: Since the sampling periods was different for the different host-species (following their oviposition activity), authors should mention as well the appearance of parasitoids in these same periods. Authors should also assure that the cold treatment of plants in order to 'delay' parasitoid and egg development does not affect the results in a species-specific manner. As well, authors should describe how (i.e. which criteria) did they evaluate egg-parasitism and egg-development; they might want to include photographs of examples.

lines 162 to 164: authors should mention the in- or dependency of the results on plant species and locality (shown in Table 4)

lines 165 to 166: authors should mention the in- or dependency of the results on locality (shown in Table 5)

line 173: the table legend reads that the data is organized by locality but in the table it is presented by species. Please, correct legend or data accordingly. Also, replace / mean for % mean

line 175: locality seems to be missing in table legend

line 180: It is not clear the locality in the figure, either include it or remove change figure legend

lines 200 to 220: the authors might want to re-write this paragraph since it lacks a bit of structure to threads ideas.

Author Response

Comments and Suggestions for Authors

In their manuscript Different oviposition strategies of closely related damselfly species as an effective defense against parasitoids, the  authors compared the proportion of parasitized eggs, and egg-development, among three species of damselfy host species. This is a nice work that with some minor changes will produce a very solid piece of work.

General comment:

authors should keep an uniform nomenclature for the species throughout the text and the figure legends (e.g. L. sponsaC. viridis and S. fusca in the text vs Lestes, Chalcolestes and Sympecma in Figure legends)

·       In methods section, where names of individual species were stated at first, full taxonomic names were stated. After that, shortened names of genus were stated. In figure legends full names were stated for clear understanding of figures

Specific comments:

lines 73 to 74: the sentence The proposed research would contribute to our knowledge of the species-specific life history traits and their influence on defense against parasitoids reads as copied from a grant application. Authors should shift the sentence to the Discussion section (e.g. these results/this study contributes to our knowledge...')

·       You are right. This sentence does not provide any additional information to text and therefore was removed from the text.

lines 77 to 84: idem comment above. If hypotheses are to be mentioned in the introduction, authors must include a reference to the validation/refusal of such hypotheses in the discussion. 

·       We are not sure, that we understand this comment. You are right that third hypothesis was speculative and we need additional experiment to test this hypothesis. Therefore we decided to remove third hypothesis.

lines 86 to 123: the information contained in the section Biology of studies species and their oviposition strategies is not mere methodological, thus authors should consider including this complete section into results.

·       We believe that biology and especially information about oviposition strategies provide very relevant context to understand all consequences. Be these information could not be stated in Results (this is based on different studies).

line 124: authors should move Table 2 to the end of Study sites section (below line 132) 

·       You are right. Thank you for suggestion.

lines 133 to 146 - Data sampling section: Since the sampling periods was different for the different host-species (following their oviposition activity), authors should mention as well the appearance of parasitoids in these same periods. Authors should also assure that the cold treatment of plants in order to 'delay' parasitoid and egg development does not affect the results in a species-specific manner. As well, authors should describe how (i.e. which criteria) did they evaluate egg-parasitism and egg-development; they might want to include photographs of examples.

·       This is an issue for Sympecma eggs, because any parasitoids were found in this time. Eggs of Lestes and Chlacolestes were sampled in one period to minimalize effect of season. All eggs in the cold were excised within 10 day, whereas according to our experience any parasitoids were unable complete development in this conditions. Photographs of parasitised, developed and un-developed eggs is stated in our study Harabis et al. 2015 – this information was improved in the current manuscript.

lines 162 to 164: authors should mention the in- or dependency of the results on plant species and locality (shown in Table 4)

·       Effect of locality was random effect, therefore is not stated in any table. Effect of oviposition plant was really absent. Thank you.

lines 165 to 166: authors should mention the in- or dependency of the results on locality (shown in Table 5)

·       Please see previous comment.

line 173: the table legend reads that the data is organized by locality but in the table it is presented by species. Please, correct legend or data accordingly. Also, replace / mean for % mean

·       It is true. Figure legend was modified.

line 175: locality seems to be missing in table legend

·       As we already stated. Effect of locality was random effect.

line 180: It is not clear the locality in the figure, either include it or remove change figure legend 

·       Legend was modified.

lines 200 to 220: the authors might want to re-write this paragraph since it lacks a bit of structure to threads ideas.

·       This paragraph was modified to be more fluent and adequate.

Reviewer 2 Report

In this study, the authors compared the proportion of parasitized eggs and undeveloped eggs of three lestid species with different oviposition strategies, focusing on their preference for specific oviposition substrate (i.e., plant species).

They found that eggs oviposited by C. viridis into very stiff substrates had a low rate of parasitoidism and the lowest proportion of undeveloped eggs compared to L. sponsa and S. fusca. Eggs oviposited by S. fusca into dead plants resulted in eggs that were complety free of parasitoids but still 14% of eggs were undeveloped.

The main question of this paper is to analyse the impact of the strategies of oviposition (preference between substrate) on the eggs (parasitoidism and survival). Nevertheless, the authors choose only one species laying on stiff substrate and one specie on “classical substrate”; in consequence, no answer can be made on this question because of confusing factors.

I really suggest the authors telling another story and just describe the results for the three different species.

The authors insist on parasitism nevertheless for S. fusca, despite no parasitoids 14% of undeveloped eggs are found. Consequently, we can deduce that paraitoidism is responsible for no more that 3% of undeveloped eggs.

In the abstract, p.1 l.22-23 supress the sentence on parental care that is hypothetical

Author Response

Comments and Suggestions for Authors

In this study, the authors compared the proportion of parasitized eggs and undeveloped eggs of three lestid species with different oviposition strategies, focusing on their preference for specific oviposition substrate (i.e., plant species).

They found that eggs oviposited by C. viridis into very stiff substrates had a low rate of parasitoidism and the lowest proportion of undeveloped eggs compared to L. sponsa and S. fusca. Eggs oviposited by S. fusca into dead plants resulted in eggs that were complety free of parasitoids but still 14% of eggs were undeveloped.

The main question of this paper is to analyse the impact of the strategies of oviposition (preference between substrate) on the eggs (parasitoidism and survival). Nevertheless, the authors choose only one species laying on stiff substrate and one specie on “classical substrate”; in consequence, no answer can be made on this question because of confusing factors.

I really suggest the authors telling another story and just describe the results for the three different species.

·       You are right that we can not talk about the benefits of the strategy, because we have only one representative for each. So in Discussion and some other sections we tried to make it clear that there are differences in mortality between species.

The authors insist on parasitism nevertheless for S. fusca, despite no parasitoids 14% of undeveloped eggs are found. Consequently, we can deduce that paraitoidism is responsible for no more that 3% of undeveloped eggs.

·       We are not sure that we understood this comment properly but in our study, we measured two things – proportion of undeveloped eggs (i.e. the pure survival of eggs in the particular substrate), and parasitism rate (the proportion of eggs attacked by parasitoids).

In the abstract, p.1 l.22-23 supress the sentence on parental care that is hypothetical

·       You are right. This sentence was removed.